# Role of Adipose Tissue microRNAs in the Onset of Metabolic Diseases and Implications in the Context of the DOHaD

**DOI:** 10.3390/cells11233711

**Published:** 2022-11-22

**Authors:** Laurent Kappeler

**Affiliations:** 1Centre de Recherche St-Antoine, CRSA, Sorbonne Université, INSERM, F-75012 Paris, France; laurent.kappeler@inserm.fr; Tel.: +33-149-284-664; 2IHU-ICAN Institute of Cardiometabolism and Nutrition, F-75013 Paris, France

**Keywords:** adipose tissue, microRNA, DOHaD, fetal programming, biomarkers, obesity, insulin resistance, diabetes

## Abstract

The worldwide epidemic of obesity is associated with numerous comorbid conditions, including metabolic diseases such as insulin resistance and diabetes, in particular. The situation is likely to worsen, as the increase in obesity rates among children will probably lead to an earlier onset and more severe course for metabolic diseases. The origin of this earlier development of obesity may lie in both behavior (changes in nutrition, physical activity, etc.) and in children’s history, as it appears to be at least partly programmed by the fetal/neonatal environment. The concept of the developmental origin of health and diseases (DOHaD), involving both organogenesis and epigenetic mechanisms, encompasses such programming. Epigenetic mechanisms include the action of microRNAs, which seem to play an important role in adipocyte functions. Interestingly, microRNAs seem to play a particular role in propagating local insulin resistance to other key organs, thereby inducing global insulin resistance and type 2 diabetes. This propagation involves the active secretion of exosomes containing microRNAs by adipocytes and adipose tissue-resident macrophages, as well as long-distance communication targeting the muscles and liver, for example. Circulating microRNAs may also be useful as biomarkers for the identification of populations at risk of subsequently developing obesity and metabolic diseases.

## 1. Introduction

### 1.1. Obesity and Metabolic Diseases

In 2016, the WHO estimated that 1.9 billion adults were overweight and that more than 650 million were obese, corresponding to 39% and 13% of the adult population of the world, respectively. These figures indicate that there is a worldwide epidemic of overweight and obesity [1]. Obesity and overweight are associated with numerous comorbid conditions, including cardio-metabolic diseases (hypertension, diabetes and insulin resistance), in particular. Indeed, obesity has been reported to increase the risk of developing type 2 diabetes (T2D) seven-fold in men and 12-fold in women in the USA and EU [2]. Diabetes was estimated to affect more than 350 million people in 2013, and the International Diabetes Federation has estimated that the number of diabetic individuals will increase to 642 million by 2040, with 90% of those affected presenting type 2 diabetes [3,4]. The WHO has estimated that diabetes already accounts for 1.6 million deaths each year [1] and 6–8% of global mortality in 20 to 79 year olds [4]. Genetics and lifestyle play crucial roles in the onset of obesity, but this epidemic is worsening as it has begun to affect children. Indeed, the WHO estimated that 340 million children and adolescents between the ages of 5 and 19 years were overweight or obese in 2016 [1]. The proportion of this age group affected rose from 4% in 1975 to more than 18% in 2016, highlighting the aggravation of the situation [1]. The earlier development of overweight and obesity may lead to an earlier onset of obesity-associated diseases and a worsening of their course.

In this context, one key issue is understanding the link between obesity and metabolic diseases can facilitate the prevention of their development. Classically measured parameters, such as glycemia, cholesterol levels and BMI, may predict the chances of T2D appearing within the next 5 to 10 years (probability of 0.85–0.9). However, these parameters are not specific to T2D and cannot be generalized to the global population [3]. Improvements to health strategies are required, and better stratification of the population to be tested is one possible way forward. By identifying the population at risk, it may be possible to intervene earlier and prevent the onset of disease. Such approaches are key elements of the precision medicine of the 21st century and the development of preventive medicine. Indeed, increasing numbers of studies are reporting that a part of the population is particularly prone to the development of obesity, and that this susceptibility may be programmed in early life. Childhood is a crucial period for strategies designed to promote healthy living, having sustained effects on health because biological systems are more “plastic” during this period and because interventions in childhood can influence responses to lifestyle factors in later life. The excess adipose tissue development that frequently precedes the onset of metabolic diseases may (i) play a crucial role in the pathophysiology of metabolic diseases and (ii) present potential levers for therapeutic actions and the identification of individuals preprogrammed to develop obesity-associated metabolic diseases.

### 1.2. Adipose Tissue

Obesity may have deleterious effects through its association with comorbid conditions, but adipose tissue is, nevertheless, crucial for life as a reserve of energy, to maintain body temperature and as an important endocrine organ. Susceptibility to the development of obesity in adulthood may be partly programmed in early life through modulation of the development of adipose tissue. Indeed, adipose tissues develop at the end of gestation to allow the fetus to adapt to life outside the womb. This phenomenon is particularly marked in humans, which have a higher percentage of adipose tissue than many other species (e.g., 15% in humans vs. 2.1% in mice and 1.3% in pigs) [5,6]. In contrast, mice and rat neonates, which are born after short gestation periods, present immature and less developed adipose tissues that fully maturate during the postnatal period [5]. Of note, the degree of maturation at birth also depends on the adipose deposit, which ranges from 0% for the epididymal tissue to 70–80% for the subcutaneous and retroperitoneal tissues in mice—although all are composed of small and immature adipocytes [7]. Once developed, the number of adipocytes remains constant in physiological conditions, with a slow turnover (around 10% per year) [8]. Adipose tissues are classically split into three categories according to the type of adipocytes they contain: white adipocytes, brown adipocytes and brite (beige) adipocytes [9]. Of Note, the different adipose deposits, and notably for the white adipose tissue, have been suggested to present different functions and regulatory settings [9]. Adipose tissues also contain mesenchymal stem cells and macrophages and are irrigated by blood vessels, which play an important role in adipose tissue pathophysiology [5,7,9,10].

Heat production through non-shivering thermogenesis, mediated by brown adipose tissue (BAT), is particularly important at birth. During postnatal development, much of the BAT is replaced by white adipose tissue (WAT), but BAT persists indefinitely at certain sites, such as clavicular, paravertebral and cervical ones, in adult humans [6]. Brite adipocytes, which are different from both white and brown adipocytes, can also produce heat. They resemble white adipocytes in basal state but express the principal characteristics of brown adipocytes in response to cold exposure for example, which allows them to produce heat. These characteristics are increased mitochondria content and UCP1 expression associated with a multilocular lipid droplet. Brite adipocytes are differentiated from white pre-adipocytes and are closer to the white adipocyte cell lineage in terms of their gene expression pattern, although it is distinct from both white and brown adipocytes [11,12]. Recent studies have reported a “beiging” capacity for certain white adipose tissues after cold exposure, such as the visceral and inguinal WAT [11,12]. Both brown and brite adipocytes are responsible for thermogenesis and can influence the “thriftiness” of metabolism later in life. However, WAT plays a particularly crucial role in the onset of obesity-associated diseases. Indeed, WAT regulates glucose and lipid storage, lipid metabolism and endocrine functions, notably through the secretion of leptin and adiponectin. WAT presents a well-known sexual dimorphism: adiposity is greater in females, who have more gynoid adipose tissues, whereas it is lower in males, who have more android adipose tissue, which is more heavily implicated in the induction of cardio-metabolic diseases [5,13]. Global WAT development, as illustrated in the omental WAT (see below), also seems to be important for determining adipocytes function and, thereby, health trajectories. WAT consists of three to four different subtypes of adipocytes originating from six to seven different types of cells [9]. The proportions of cell types differ in each adipose deposit (e.g., pericardial versus the inguinal deposits, etc.) [9], and variations due to developmental selection may modify their regulations. In addition, it has recently been reported for omental WAT that pre-adipocytes with high or low levels of CD9^+^ labeling can differentiate into mature adipocytes with more fibrotic and pro-inflammatory or more adipogenic phenotypes, respectively [14]. These data highlight the importance of the developmental window in the programming of adipocyte function. Modification of the nutritional/endocrine environment during the fetal/early-postnatal period may, therefore, alter the development of adipose tissue and promote adiposity, together with the subsequent development of associated metabolic diseases, through a mechanism using the developmental origin of health and adult diseases (DOHaD) concept.

## 2. The Developmental Origin of Health and Adult Diseases

### 2.1. The DOHaD Concept

Chronic non-communicable diseases appear to result from a combination of genetic, environmental and behavioral factors. Studies have consistently shown that the life history of children plays a key role in the development of cardio-metabolic diseases later in life (see [15]). Indeed, environmental changes during both the fetal and early postnatal periods are crucial for the programming of cardio-metabolic diseases in adults. The results of many studies on cohorts of individuals born small for gestational age [16] and human populations exposed to famine (e.g., hunger cohorts in the Netherlands, Ukraine, Russia, China, Bangladesh, etc.) [17,18] have supported the concept of a developmental origin of health and adult diseases [19]. According to this concept, modifications may favor adaptation to an anticipated future environment, increasing the chances of survival. However, if the real environment differs from that anticipated, the body will be more sensitive and prone to diseases development. The modifications involved may include alterations of organogenesis induced directly by environmental stimuli during the developmental period and functional modulations leading to a new regulatory setup.

The DOHaD concept encompasses almost all the physiological functions of the organism, including metabolic functions, in particular. An association of adiposity with metabolic diseases has been observed in human populations. Healthy pregnant women with a healthy diet have babies with lower levels of adiposity, and breastfed newborns and newborns fed with low-protein formula milk have smaller fat mass gains at the age of five to eight years [20]. In contrast, maternal obesity and a high-protein diet during the first two years of life are associated with a high BMI at nine years of age and in adulthood, with greater adiposity and a higher IR in adipose tissues [21,22,23,24]. The direct fetal/neonatal environment and the modulation of key-nutrient/toxin exposure, which mainly involves the maternal environment, are not the only causes of these modifications. Epigenetic regulations are also involved and imply potential indirect inputs, as suggested by associations with paternal history [25,26,27,28,29].

### 2.2. Populations Affected by the DOHaD and Preclinical Study Models

Many factors and critical periods have been suspected to influence the developmental trajectory, giving rise to a similar number of preclinical models [15]. The factors and models considered to date include a deficit of oxygen plus nutrients (as a model of pre-eclampsia) [30], glucocorticoids (stress exposure) [31], endocrine disruptors and toxins (exposure to pollutants, smoking) [32], inflammatory factors (infectious diseases) [33] and nutritional deficit or overload (de- or malnutrition/gestational diabetes/parental obesity) [34]. Globally, the entire development period appears to be pertinent to DOHaD, extending from the preconception period, in which conditions can influence gamete maturation, right up to the end of development, with the phenotype observed depending on the periods of proliferation/differentiation of each organ. The end of the critical period remains a matter of discussion but is thought to lie about 1000 days after conception for most phenotypes in humans, although some reports suggest that it may lie towards the end of the teenage–adult period for behavioral and mental health susceptibilities [35,36].

The modulation of nutrition is one of the most powerful levers acting on the programming of obesity and subsequent associated comorbidities in offspring. Interestingly, the risk of developing obesity and related diseases in response to nutritional alterations follows a U-shaped curve. At one end of the curve, maternal obesity and overweight and associated gestational diabetes are associated with the programming of obesity in offspring [37,38]. The preconception period has also been identified as important for the metabolic programming of offspring [39]. However, the direct fetal environment (i.e., the maternal environment) is not the only factor involved in the programming of obesity in offspring. Indeed, the current obesity pandemic has highlighted the role of paternal obesity [40,41]. At the other end of the curve, being born small for gestational age (SGA) is also associated with cardio-metabolic diseases [16]. In particular, the frequency of cardio-metabolic diseases in adulthood is higher in individuals born with intra-uterine growth retardation (IUGR) than in those born with a normal weight [17,42]. SGA affects 5–7% of live births in developed countries [43], increasing to 27% of all live births in low- and middle-income countries, with a prevalence of 10% to >75% in Nepal and India [44,45,46,47,48]. The vast majority of infants born SGA or with IUGR display catch-up growth and reach the normal growth curve during the first few years of life [49,50]. The early postnatal period also seems to be a window for hypersensitivity for the programming of obesity. Indeed, the development of obesity in adults has been associated with excess weight gain in the first six months of life [51]. Moreover, rapid catch-up growth in babies born SGA is associated with obesity and earlier onset of metabolic diseases compared to infants with late catch-up growth [52]. The population of infants born SGA or with IUGR is probably fueling the worldwide epidemic of obesity and metabolic syndrome. These individuals may be one of the first populations that should be tested for the risk of developing obesity and associated metabolic diseases.

### 2.3. Mechanisms of the DOHaD

#### 2.3.1. Changes in Organogenesis

Environmental insults during the developmental period may have direct impacts on organogenesis, and the impact of fetal undernutrition on glomeruli in the kidney and on the numbers of β-cells in the pancreas is well-known [15]. Each organ develops along a specific timeline and may, thus, present a specific window of hypersensitivity. In addition, genetics and the speed of development may also influence the sensitivity of the fetus to changes in the new environment. Indeed, many studies have highlighted the sexual dimorphism of fetal programming. The rate of embryonic cell division and metabolism have been reported to be slower in female fetuses than in male fetuses, potentially resulting in differences in sensitivity to environmental fluctuations [13,53]. Moreover, some studies have associated this sexual dimorphism in fetal programming with the sexual dimorphism in placental function, including, in particular, the ability of the placenta to adapt to nutritional changes [54,55].

Adipose tissues develop towards the end of the fetal development period, and alterations to the development of these tissues may have an impact on subsequent health trajectory. Moreover, the sexual dimorphism of adiposity in newborns indicates developmental differences that may also modulate the sensitivity of male and female infants to the development of obesity later in life [13,53]. BAT development plays an important role in determining the likelihood of subsequent obesity development through its role in regulating thermogenesis and energy expenditure (e.g., resulting in different degrees of metabolic thriftiness) [6,56,57,58]. Alterations to WAT development may also be involved. Indeed, WAT may increase its storage capacity in response to chronic energy overload through hyperplasia and/or hypertrophy, the latter of which has been reported to promote the occurrence of metabolic diseases [59,60]. This balance between WAT hyperplasia and hypertrophy may involve glucocorticoid signaling, which is known to summarize almost all exposures for fetal programming [59,61,62,63]. Moreover, it has also been suggested that the modulation of the fetal/neonatal environment during the developmental period may lead to the selection of different adipocyte subtypes, potentially affecting the subsequent reactivity of the adipose tissue. Such selection has been reported for two platelet-derived growth factor receptor-α-positive (PDGFRα^+^) WAT pre-adipocyte subtypes in the omental WAT, one with high levels of CD9 expression and the other with low levels of CD9 expression [14]. The development of these white adipocytes leads to more fibrotic and proinflammatory (PDGFRα^+^/high-CD9^+^) or more adipogenic phenotypes (PDGFRα^+^/low-CD9^+^). In humans, the frequency of PDGFRα^+^/high-CD9^+^ progenitors in omental WAT has been associated with IR severity and type 2 diabetes [14].

#### 2.3.2. Epigenetic Regulation

Independently of the consequences of the developmental alterations described above, the nutritional environment during the pre- and postnatal periods may trigger permanent functional deregulations. The new functional equilibrium is established through epigenetic regulation that can be maintained over very long periods and through cell divisions to regulate gene expression [15,64]. Epigenetics is a fundamental mechanism of cell specialization throughout development. Epigenetics also plays a crucial role in the adaptation of the organism to its environment. In particular, adaptation of the organism to the capacity of a given environment to supply nutrients is essential for its survival. Furthermore, as epigenetic mechanisms involve donors of methyl or acetyl groups and vitamins, epigenetic modulation is highly dependent on nutrition.

Epigenetic modifications regulate gene transcription through mechanisms such as the methylation of DNA at CpG sites and posttranslational modifications of histone tails. DNA methylation at CpG sites is generally associated with an inhibition of gene expression, whereas histone modifications may either upregulate or downregulate gene transcription [65]. Indeed, various types of histone modifications may occur (e.g., methylation, acetylation, phosphorylation, ubiquitination, etc.), the effects of which depend on the number and type of modifications and the residue modified [66]. Epigenetic mechanisms can also act at the posttranscriptional level. This regulation involves microRNAs and is generally associated with an inhibition of protein translation. The impacts of DNA methylation and histone modifications have been studied in the context of mesenteric adipose tissue programming [51,67,68,69]. Such epigenetic modifications have a particularly marked effect in people born with IUGR or SGA, who frequently present cardio-metabolic diseases as adults [15]. The role of microRNAs in adipocyte function has also recently attracted attention, as shown by a recent review [70]. The findings generated by these new avenues of research suggest that adipose tissue may use microRNAs in an original way, to induce metabolic alterations in other key metabolic organs, thereby promoting the onset of metabolic disease.

## 3. MicroRNAs

### 3.1. MicroRNA Biogenesis

Only a small part (about 2%) of the human genome is thought to encode proteins [71]. A large proportion of the transcribed genome instead corresponds to non-coding RNAs (ncRNAs). These ncRNAs nevertheless have a function, playing crucial roles in cell biology. They include ribosomal (rRNA), transfer (tRNA), micro (mir), small nucleolar (snoRNA) and long non-coding (lncRNA) RNAs. MicroRNAs are a type of ncRNA of particular interest (Figure 1).

MicroRNAs are single-stranded, 21–25 nucleotide-long ncRNAs that are widely expressed and play a fundamental role in the posttranscriptional regulation of gene expression. The main pathways of microRNA biogenesis involve the transcription of a pri-microRNA from a non-coding gene by the type 2 polymerase. The synthesis of pri-microRNA is dependent on active promoters and is, thus, subject to regulation by classical external effectors and transcription factors. For example, the synthesis of mir-155 is directly stimulated by LDL [4] and the synthesis of mir-29a and mir-122 is stimulated by high levels of exposure to glucose [72]. The promoters of microRNA-encoding genes can also be subject to epigenetic regulation (Figure 1). For instance, a decrease in levels of the trimethylated-lysine 4 modification of histone H3 (H3K4me3) has been observed in the promoter of the pri-mir-17-92 gene in the liver of adult mice born with IUGR, this modification being associated with lower levels of pri-mir-17-92 RNA [73]. Consistently, levels of mir-19a-3p, which is also encoded by pri-mir-17-92, were also lower in these mice.

Once transcribed, pri-microRNAs adopt a hairpin configuration that is recognized by Drosha and DiGeorge syndrome critical region protein 8 (DGCR8) proteins. DGCR8 is a double-stranded RNA-binding protein and Drosha is a type 3 endoribonuclease that cleaves the pri-microRNA approximately 22 base pairs away from the loop to generate the pre-microRNA. Alternatively, Drosha may process pre-microRNA directly via a non-canonical pathway cleaving small introns (mirtrons) generated from the splicing of protein-coding mRNAs [74]. In both cases, the pre-microRNAs are exported to the cytoplasm for further processing. The canonical pathway involves processing by a protein complex containing Dicer and TRBP (Tar RNA binding protein). This complex removes the hairpin and generates a microRNA duplex, with each strand presenting two extra bases at its 3′ end. The active strand, originating from the 5′ (-5p) or 3′ (-3p) part of the stem loop, is delivered to the RNA-induced silencing complex (RISC) by the Argonaute 2 (Ago2) protein. Parallel to this pathway, the Dicer-independent processing of microRNAs by Ago2 has also been reported [75].

The mature microRNAs contain a seed sequence of six to eight nucleotides at their 5′ end. This sequence principally recognizes the 3′UTR of the targeted mRNA(s). Nucleotides at the 3′ end can also bind the mRNA, but the inhibition of translation is driven by the seed sequence. The presence of a mismatch within the seed sequence leads to the inhibition of translation, whereas a perfect match leads to deadenylation followed by degradation of the mRNA (Figure 1).

About 2600 human microRNAs have been identified in the miRBase database (v22) [76]. Given the relatively short length of the seed sequence, a single microRNA can target many mRNAs and, conversely, a single mRNA can be targeted by numerous microRNAs. Moreover, 70% of mRNAs are thought to have sites for microRNAs from multiple families, with a mean of 4.2 microRNA-binding sites per targeted 3′UTR [77], enabling the cooperative regulation of mRNA translation by microRNAs. All organs contain microRNAs. Some of these microRNAs, such as let-7, are ubiquitously expressed, whereas others are more abundant in some organs than in others, such as mir-122 in the liver or mir-375 in the pancreas [78]. Moreover, it has been reported that the same microRNA can have different roles in different organs, consistent with its many possible targets. For example, mir-33a has been reported to regulate cholesterol and lipid metabolism in the liver and the M1 vs. M2 polarization of macrophages [78].

### 3.2. Role of microRNAs in Adipose Tissue Development and Pathophysiology

The first analyses of the mechanisms associated with hypersensitivity to the development of severe obesity focused on genetic approaches and highlighted numerous loci in humans. Interestingly, more than 200 of these loci coincide with the sites of microRNA-coding pseudogenes [79]. In human subcutaneous adipose tissue, more than 600 microRNAs have been identified as expressed and predicted to target 30% of protein-coding mRNAs [80], suggesting a key role in the development and function of adipocytes [70,81].

#### 3.2.1. Role of microRNAs in Adipose Tissue Development

MicroRNAs modify the development of BAT in the offspring of obese female mice, establishing a thrifty metabolism favoring the development of obesity later in life. This alteration involves an increase in the abundance of mir-199a-5p/-501-5p/-383-3p, which are known to suppress BAT adipogenesis in E_18.5_ fetuses, associated with a decrease in the biogenesis of mitochondria crucial for BAT differentiation [58]. Consistently, constitutive mir-155 invalidation has been shown to be associated with an increase in the differentiation of brown adipocytes, increasing heat production and preventing the induction of obesity by a high-fat diet [56]. WAT levels were slightly lower in KO mice and white adipocyte hypertrophy was prevented in animals fed a high-fat diet, thereby preventing local inflammation. Numerous microRNAs seem to be involved in WAT differentiation, particularly those synthesized via the canonical pathway, as illustrated by the impact of Dicer KO on WAT development in mice, notably the subcutaneous and intra-abdominal WAT [82]. Indeed, in addition to the abovementioned microRNAs, other studies have revealed roles for mir-150/-141/-143/-200a-c/-204/-429 in the differentiation of BAT and/or WAT [80,82]. Nutritional restriction alters mir-483-3p levels in the epididymal adipose tissue of adult men who were born SGA [83]. Such alterations were observed in the offspring of mothers fed a low-protein diet throughout pregnancy and suckling [83]. It is thought that mir-483-3p inhibits adipocyte differentiation at least partially through the targeting of growth differentiation factor 3 (GDF3). This microRNA may also affect adipocyte expandability through its effects on GDF3 synthesis, favoring ectopic lipid accumulation and the occurrence of metabolic diseases.

As suggested for mir-483-3p, the actions of microRNAs may be complicated by the ability of some of these molecules to modulate both adipocyte ontogeny and function. This is the case, for example, for the pri-mir17-92 pseudogene, which encodes five different microRNAs. The 3T3 L1 pre-adipocyte proliferation is modulated by the inhibitory effects of pri-mir17-92 on rb2p130, which is required for adipose tissue development, and insulin signaling sensitivity is modulated by the action of pri-mir17-92 on PTEN [80,82,84].

#### 3.2.2. Role of microRNAs in Adipose Tissue Functions

As key players in adipogenesis and adipocyte function, the PPAR-ɣ and C/EBP-β mRNAs seem to be frequent targets of microRNAs in adipocyte programming, as reported for mir-130b/-27a/-27b/-130/-378-3p/-374b [82,85,86,87]. In particular, it has been suggested that mir-27a induces IR by inhibiting PPAR-ɣ, and that its levels increase in prediabetic individuals with obesity [88]. This microRNA will be discussed further in Section 3.3.

Similarly, the Irs1 mRNA plays an important role in the development of IR and is a frequent target of microRNAs in adipocytes: IRS_1_ regulates insulin-dependent glucose uptake by GLUT_4_ and alterations in its levels can favor the onset of IR. The levels of mir-222 increase in a dose-dependent manner in response to treatment with high concentrations of glucose in 3T3 L1 cells and this microRNA has been reported to promote IR by directly inhibiting IRS-1 translation, thereby preventing Glut4-mediated glucose uptake [9,72]. Levels of mir-222 have been correlated with increases in BMI between the ages of 7 and 10 years in children of normal weight [89], a sign indicative of potential obesity development later in life. A similar inhibition of IRS-1 translation was also achieved with mir-126-3p in the epididymal fat pads of mice with high-fat diet (HFD)-induced obesity [90]. This regulation was subsequently confirmed in vitro, with primary adipocytes differentiated from pre-adipocytes harvested from the offspring of obese mice with sustained high levels of mir-126-3p [91]. Furthermore, mir-126-3p also plays a role in ER stress by inhibiting lunapark [91], a combinatorial effect reproduced by mimic transfection [92]. This deregulation induced by mir-126-3p may highlight an early mechanism accounting for the risk developing type-2 diabetes later in life in the offspring of obese dams.

Globally, these data indicate that various regulators of the insulin signaling pathway may be targeted in parallel by microRNAs to induce IR. For example, mir-29a, the levels of which are increased by treatment with high concentrations of glucose in 3T3 L1 cells, has been reported to induce IR, partly by inhibiting secreted protein acidic rich in cysteine (SPARC) [72,93,94]. A meta-analysis of studies published between 1993 and 2014 highlighted an increase in mir-29a levels in the adipose tissues of patients with T2D and obese/insulin-resistant animal models, together with increases in mir-34a/-142-3p/-106b/-107 levels [95]. Increasing local inflammation in adipose tissue is another important aspect for IR development that may also be associated with microRNA alterations. For example, a decrease in mir-706 levels has been observed in the epididymal adipose tissue of the offspring of obese dams and associated with increases in CMK1D and IL33, two predicted inflammation-related targets [21]. These modifications may occur early in life, as reported for mir-181a and mir-221, which are involved in regulating inflammation and the levels of which are already low in neonates born to mothers with pregestational obesity [96].

Globally, these data indicate that microRNAs may be involved in adipogenesis and/or adipocyte function, and that their levels change during obesity and metabolic diseases, such as IR. The alterations to the levels of these microRNAs in early life suggest a role in the programming of metabolic alterations in adipose tissues, facilitating the onset of global insulin resistance and T2D. The progression of local IR from adipose tissues to other key metabolic organs is a crucial aspect of the induction of metabolic diseases in which microRNAs may play a key role.

### 3.3. Role of Adipose Tissue microRNAs in Communication between Organs

#### 3.3.1. Role of microRNAs from Adipocytes

MicroRNAs regulate protein synthesis by modulating protein translation in the cell cytoplasm but are highly abundant in all fluids of the body [97]. The presence of microRNAs in fluids is variable, with around 200 different microRNAs found in urine but more than 400 present in breast milk, seminal fluid and saliva. The global concentration of microRNAs is also highly variable, from 90 µg/L in urine to almost 50 mg/L in breast milk. In plasma, about 350 different microRNAs have been identified, with a global concentration of about 300 µg/L [97]. They are associated with lipoprotein (e.g., HDL), apoptotic bodies and RNA-binding proteins (e.g., Ago2), or are present in microvesicles or exosomes [78]. Microvesicles bud from the membrane and are 100–1000 nm in diameter, whereas exosomes are formed by the fusion of multivesicular bodies (MVB) with the plasma membrane and are 40–100 nm in diameter (Figure 1). Exosome are also characterized by specific surface markers, such as CD63, and their secretion can be stimulated by ceramides, glucose, free fatty acids, TGF-β or H_2_O_2_ [98,99,100,101,102,103].

Recent studies have indicated that adipocytes may be a major source of exosome-associated microRNAs. Indeed, the selective deletion of DICER in mouse adipocytes (Adipo-DICER-KO mice) has been shown to decrease the amounts of microRNA in extracellular vesicles [104]. Similar decreases were also observed following the selective deletion of DGCR8 from mouse adipocytes (Adipo-DGCR8-KO mice) [105]. Moreover, a specific modulation of microRNAs may occur in the exosomes. Indeed, specific changes in the levels of exosome-associated microRNAs relative to HDL-associated microRNAs has been reported following bariatric surgery in humans [106]. These data suggest that the release of exosomes and their loading with microRNAs may be regulated. Consistent with such a mechanism, one recent study showed that cells may have different microRNAs profiles in their exosomes and cytoplasm [107]. This selection may be based on a small 3′ sequence and may involve RNA-binding proteins (RNABPs). The multiple motifs carried by microRNAs are thought to play a major role in modulating their sorting [107]. Such regulation indicates the occurrence of an active process and suggests a specific role for microRNAs in the bloodstream. Consistently, recent reports have suggested that circulating microRNAs may influence the emergence of metabolic diseases by regulating organs at a distance. The mir-27a/-34a/-141-3p/-155/-210/-222 microRNAs are frequently observed and have been suggested to play a role in IR development, in particular [99].

The mir-27a-3p microRNA is of particular interest. It originates from chromosome 8 and is predicted to target PPAR-ɣ and PPAR-α [108,109]. It has a seed sequence similar (one nucleotide different) to that of mir-27b-3p, which originates from chromosome 13. Various tissues express mir-27a-3p, and the expression of this microRNA by mature adipocytes may be involved in IR development and interorgan communication. In the context of obesity, mir-27a-3p levels in exosomes increase [88,110]. The injection of exosomes isolated from obese mice and containing mir-27a-3p into lean mice is sufficient to induce glucose intolerance and IR [110]. Exosomes have been shown to target epididymal WAT and the liver, in which mir-27a-3p/-27b-3p appear to target PPAR-α and PPAR-ɣ [110,111,112]. Furthermore, the exosomes secreted by adipocytes accumulate in muscle, in which they prevent glucose uptake by directly inhibiting PPAR-ɣ [88]. Interestingly, mir-27a-3p levels are high in pre-diabetic obese individuals, suggesting possible deregulation in the early phase of disease and a potential for use as a biomarker.

Higher levels of mir-141-3p have also been found in the exosomes circulating in the blood of obese mice [113], and it is thought that they may have effects in muscle. Human mir-141-3p is transcribed from chromosome 12 and is predicted to target the Pten mRNA. The exposure of AML-12 myocytes cultured in vitro to exosomes isolated from obese mice leads to an impairment of AKT phosphorylation by insulin. The expression of a mir-141-3p mimic by adipocytes leads to an increase in mir-141-3p levels in adipocyte-derived exosomes and a decrease in PTEN levels in AML-12 myocytes exposed to these exosomes [113].

Muscles are not the only target of the exosomes released by adipocytes. mir-222-3p is expressed from the X chromosome in humans, and its processing appears to require DICER activity in adipocytes, as suggested by studies in the Adipo-Dicer-KO mouse model. With obesity, mir-222-3p levels increase in the epididymal WAT and circulating exosomes [114]. The authors of this study used GFP-tagged CD63 expression on adipocytes to follow the resulting fluorescent exosomes in blood and revealed an accumulation of these exosomes in the muscles and liver. Hepatocyte targeting was confirmed in vitro with the HEPA 1-6 hepatocyte cell line, in which mir-222-3p altered AKT phosphorylation by inhibiting translation of the Irs1 mRNA. These effects were further confirmed in vitro with luciferase assays, as well as by both mimic and antagomir transfection in vitro and in vivo [114].

These studies highlight the role of the exosomes secreted by adipocytes in the propagation of metabolic deregulation and the onset of metabolic diseases. It has been suggested that these exosomes target various organs, with target selection probably involving differences in the surface proteins carried by the exosomes [104,115]. These targets may be located some distance away (e.g., muscles, liver) or may be local (e.g., pre-adipocytes, resident macrophages). The important role of resident macrophages in promoting the onset of metabolic diseases has been highlighted in recent years by studies showing the stimulation of inflammation by M1 macrophages and the protective effects of M2 macrophages. One recent study reported an important role for microRNAs in polarizing macrophages to the M1 or M2 phenotype [81]. For example, it was suggested that mir-155 and mir-27a modulate macrophage infiltration/activation and that mir-34a influences M1 polarization, potentially by targeting the KLF4 mRNA [81,99,116].

#### 3.3.2. Role of microRNAs from Adipose Tissue-Resident Macrophages

The macrophages resident in adipose tissue have been implicated in the propagation of metabolic diseases through modulation of inflammation. Recent studies have also highlighted an important role in micro-RNA-mediated regulation. The resident macrophages of the adipose tissue have been shown to exert paracrine effects on neighboring adipocytes and long-distance effects on muscle, pancreatic β-cells and hepatocytes [81].

The transfer of microRNAs from macrophages to adipocytes has been demonstrated in Transwell co-cultures in vitro with the expression a Cy3-tagged mir-223 by macrophages from visceral WAT [117]. Exosomes containing these microRNAs were targeted at the muscles and liver, leading to a decrease in insulin sensitivity, as confirmed by in vivo and in vitro approaches. Ting Liu et al. demonstrated the transfer of mir-29a from the visceral adipose tissue-resident macrophages to adipocytes, myocytes and hepatocytes via exosomes [2]. Luciferase assays have shown that mir-29a directly targets the PPAR-ɣ mRNA, and its role in IR induction was further confirmed with mimic- and antagomir-based experiments. Interestingly, resident macrophages may also mediate opposite patterns of regulation. Indeed, the injection into obese mice of exosomes isolated from lean mice improved the insulin sensitivity in the obese mice, and the transplantation of mir-155 KO macrophages into WT mice subsequently subjected to a high-fat diet decreased glucose intolerance and IR in these mice [117].

All these data are consistent with a role for microRNAs in the long-distance regulation of other tissues (Figure 2). They also suggest that adipocytes and adipose tissue-resident macrophages may be one of the principal sources of exosome microRNA in the blood, modulating the onset of IR in other key metabolic organs and, thus, metabolic diseases.

## 4. Potential Use of microRNAs as Biomarkers for Metabolic Diseases

The secretion of microRNAs into the blood can reveal low-grade alterations in key metabolic organs. Indeed, microRNAs have been reported to be highly resistant to degradation [118]. Accordingly, microRNAs are already beginning to be used for patient stratification for the choice of adjuvant therapies for certain cancers [119,120]. This use of microRNAs has already been initiated for metabolic alterations [121]. For instance, it has been suggested that mir-34a could be used in diagnosis to distinguish between individuals suffering from non-alcoholic fatty liver disease (NAFLD) and those with non-alcoholic steatohepatitis (NASH) [122]. Interestingly, microRNAs have also recently been proposed as early biomarkers to enhance predictions of the course of NAFLD in patients [123]. Similarly, microRNAs involved in early stages of type 2 diabetes, such as mir-184 and mir-124a, which are associated with beta-cell proliferation and differentiation, have been proposed as possible biomarkers [124]. Furthermore, changes in mir-375 levels in the blood have been reported five years before the appearance of prediabetes and type 2 diabetes in studies of human populations [125,126]. Finally, the decrease in microRNA-19a-3p levels in the blood of young adult (3 month old) male mice was associated with the emergence of insulin resistance during middle age (12 month old mice) in a mouse model of IUGR [73]. This work with a preclinical model suggests that biomarkers may be detectable nine months before IR onset in mice (a time period equivalent to decades in humans). These data are consistent with the idea that cardio-metabolic diseases are programmed in the fetus, with alterations induced but remaining silent for long periods before the emergence of the first phenotypes. MicroRNAs may, therefore, provide us with opportunities for assessing early biomarkers of the risk of developing metabolic diseases later in life, opening up new possibilities for personalized precision medicine to help people before the onset of metabolic disease. The microRNAs originating from adipose tissue are a potential target that is of particular interest in this context.

## Figures and Tables

**Figure 1 cells-11-03711-f001:**
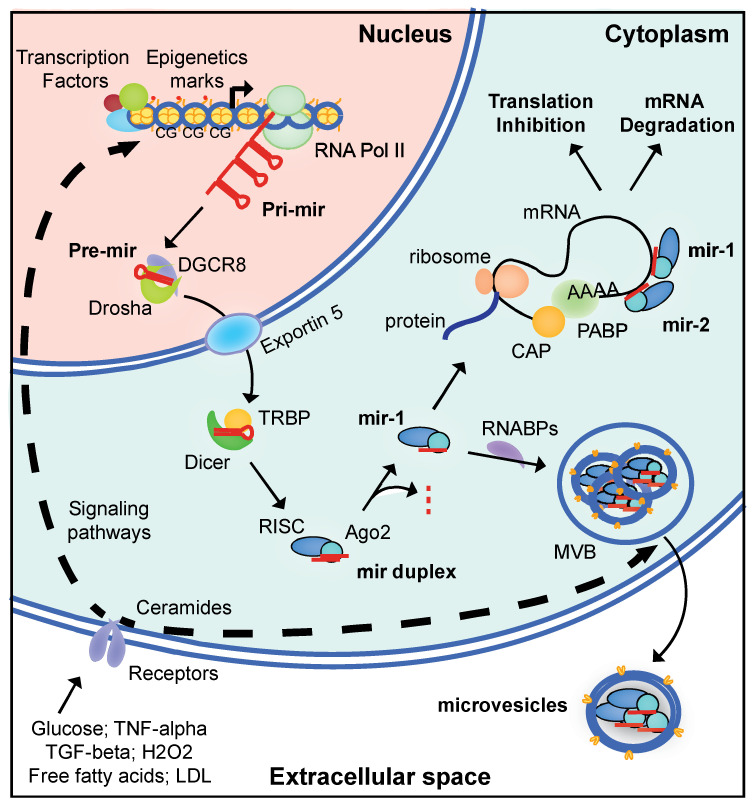
Canonical pathways of biosynthesis, mechanism of action and secretion by adipocytes of microRNAs. Following external stimulation, non-coding genes are transcribed by the RNA polymerase II (RNA pol II) to produce a pri-microRNA (pri-mir). These genes are under classical transcriptional regulation by transcription factors and epigenetics marks, such as DNA methylation at CpG sites (CG) and histone post-translational modifications. The pri-mir is processed by Drosha and DGCR8 to produce the pre-microRNA (pre-mir), which is transferred to the cytoplasm for further processing by Dicer and TRBP. The cleaved mir duplex is carried by Ago2 into the RISC complex, and the active mir strand of 21–25 nucleotides is driven to the 3′UTR of the targeted mRNA(s). Depending on the presence of a mismatch between the seed sequence of the mir and the mir-binding site of the mRNA, mRNA translation is inhibited or the mRNA is degraded. In parallel, mir can be transferred by RNA binding proteins (RNABPs) into multivesicular bodies (MVB) for exosome secretion (CD63^+^). These exosomes can then target cells at long distance.

**Figure 2 cells-11-03711-f002:**
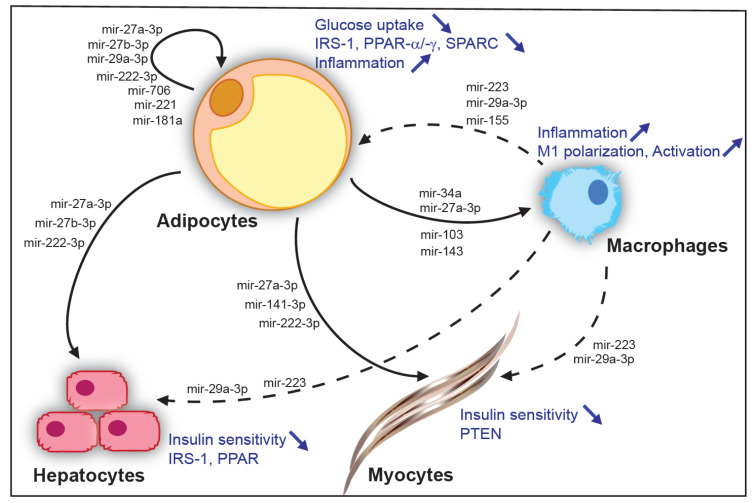
Adipose tissue microRNAs in communication between organs. Examples of microRNAs secreted by adipocytes of white adipose tissue and their local action on neighbor adipocytes and resident macrophages of the adipose tissue or, at long distance, on hepatocytes and muscles (solid arrows). Examples of microRNAs originating from resident macrophages of the adipose tissue and their local action on neighbor adipocytes or, at long distance, on hepatocytes and muscles (dashed arrows). Effects on cells types are indicated in blue.

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
