# Peer review of "Role of Adipose Tissue microRNAs in the Onset of Metabolic Diseases and Implications in the Context of the DOHaD"

_cells, 2022, doi:10.3390/cells11233711_

Round 1

Reviewer 1 Report

The review submitted for consideration in Cells present intriguing concept of the link between microRNAs and metabolic diseases. The data is well structured and analyzed. In my opinion this review will be of great interest for the auditory of Cells and the obesity research community. Few minor points have to be addressed:

1) Figure 1 is very good and descriptive.
Figure 2 could be improved with not only marking the selected examples
of microRNAs but to tag the effect that is produced in the different cells types.
2) The reference formatting have to be revised according to the Journal guidelines.

Reviewer 2 Report

Review comments

In this manuscript, the author provided a detailed summary of the currently available data regarding the role of microRNA in the regulation of adipose tissue development and function. The author further provided a rationale that, based on the summarized studies, microRNAs could be used as biomarkers for identifying metabolic dysfunction associated with the DOHaD. Overall, the manuscript flows well and is easy to follow, and the studies summarized are recent. However, the section describing adipose tissue biology was not as well constructed as to the section describing microRNA, which needs some clarification. Specifically, the types of adipose tissues used in these studies were not always included in author’s summaries. In some instances, the author summarized the studies but did not reference the corresponding papers at the end of sentences. Some editorial issues need to be corrected prior to publication, such as inconsistent font size in some sections that needs to be corrected prior to publication. Lastly, some of the studies summarized in this manuscript did not involve testing the direct association between microRNAs and DOHaD. The author needs to modify the title to better describing the content of the manuscript. Following are the review comments.

1. Adipose tissues function not only to serve as reserves of energy and to maintain body temperature, but they also are important endocrine organs. Please include this at the end of sentence (pg2, line 68).

2. Please list the type of species that have short gestation periods and provide references at the end of this statement (pg2, line 75)

3. “………. with a slow turnover.” Please provide a reference for this sentence (pg2, line 76)

4. “Adipose tissues are classically split into three categories according……” Please provide references for the type of adipose tissues at the end of the sentence (pg 2 line 79).

5. “ ………which play an important role adipose tissue pathophysiology.” Please provide references at the end of this sentence.

6. “ ………. but BAT persists indefinitely at certain sites” Which sites? (pg 2, line 84)

7. “ …..which are different from both white and brown adipocytes, can also produce heat.” What are the differences between beige and brown adipocytes?

8. “Recent studies have reported a “beiging” ……….” Reference list is needed for this sentence. (pg. 2 line 85)

9. “WAT development ………” Which type of WAT? (pg.2 line 93)

10. “Studies have consistently shown that the ……….” Reference list is needed for this sentence (pg3, line 107)

11. “The factors and models considered to date include …….” Please provide reference list for the factors described in this sentence. (pg.3 line 138)

12. “The end of the critical period remains a matter of discussion, but is thought to lie about 1000 days ……….” Please provide a reference for this statement. (pg3, line 142)

13. “BAT development plays an important role ………..” Please provide a reference at the end of this sentence. (pg.4, line 189)

14. “Indeed, WAT 192 may increase its storage capacity in response to chronic energy overload,……..” Please provide a reference at the end of this sentence.

15. “In humans, the frequency of PDG-204 FRα+/high-CD9+ progenitors in omental WAT ………..” Please provide a reference at the end of sentence. (pg.5, line 204)

16. “The impacts of DNA methylation and histone modifications have been extensively studied ………” Please include references at the end of this sentence. (pg. 5, line 226)

17. “Only a small part (about 2%) of the human genome is thought to encode proteins.” Please provide a reference at the end of this sentence. (pg5, line 237).

18. “Pre-adipocyte proliferation is modulated by the inhibitory effects of 319 pri-mir17-92 on rb2p130, ……” Please indicate which type of pre-adipocyte in the text. (pg. 7, line 319)

19. The font size and font effects are not consistent in some sections of the manuscript.

Round 2

Reviewer 2 Report

The author addressed all my comments. However, the revised manuscript needs some editing to correct typos and grammar errors.